# A Stable and Scalable Method for Solving Initial Value PDEs with Neural Networks

**Marc Finzi**[1]*, **Andres Potapczynski**[1]*, **Matthew Choptuik**[2], **Andrew Gordon Wilson**[1]
New York University[1] and University of British Columbia[2]

## Abstract

Unlike conventional grid and mesh based methods for solving partial differential equations (PDEs), neural networks have the potential to break the curse of dimensionality, providing approximate solutions to problems where using classical solvers is difficult or impossible. While global minimization of the PDE residual over the network parameters works well for boundary value problems, catastrophic forgetting impairs applicability to initial value problems (IVPs). In an alternative *local-in-time* approach, the optimization problem can be converted into an ordinary differential equation (ODE) on the network parameters and the solution propagated forward in time; however, we demonstrate that current methods based on this approach suffer from two key issues. First, following the ODE produces an uncontrolled growth in the conditioning of the problem, ultimately leading to unacceptably large numerical errors. Second, as the ODE methods scale cubically with the number of model parameters, they are restricted to small neural networks, significantly limiting their ability to represent intricate PDE initial conditions and solutions. Building on these insights, we develop Neural-IVP, an ODE based IVP solver which prevents the network from getting ill-conditioned and runs in time linear in the number of parameters, enabling us to evolve the dynamics of challenging PDEs with neural networks.

## 1 Introduction

Partial differential equations (PDEs) are needed to describe many phenomena in the natural sciences. PDEs that model complex phenomena cannot be solved analytically and many numerical techniques are used to computer their solutions. Classical techniques such as finite differences rely on grids and provide efficient and accurate solutions when the dimensionality is low ($d = 1, 2$). Yet, the computational and memory costs of using grids or meshes scales exponentially with the dimension, making it extremely challenging to solve PDEs accurately in more than 3 dimensions.

Neural networks have shown considerable success in modeling and reconstructing functions on high-dimensional structured data such as images or text, but also for unstructured tabular data and spatial functions. Neural networks sidestep the "curse of dimensionality" by learning representations of the data that enables them to perform efficiently. In this respect, neural networks have similar benefits and drawbacks as Monte Carlo methods. The approximation error $\epsilon$ converges at a rate $\epsilon \propto 1/\sqrt{n}$ from statistical fluctuations where $n$ is the number of data points or Monte Carlo samples. Expressed inversely, we would need: $n \propto e^{2\log 1/\epsilon}$ samples to get error $\epsilon$, a compute grows exponentially in the number of significant digits instead of exponential in the dimension as it is for grids. For many problems this tradeoff is favorable and an approximate solution is much better than no solution.

Thus, it is natural to consider neural networks for solving PDEs whose dimensionality makes standard approaches intractable. While first investigated in Dissanayake & Phan-Thien (1994) and Lagaris et al. (1998), recent developments by Yu et al. (2018) and Sirignano & Spiliopoulos (2018) have shown that neural networks can successfully approximate the solution by forcing them to satisfy the dynamics of the PDE on collocation points in the spatio-temporal domain. In particular, the global collocation approaches have proven effective for solving boundary value problems where the neural network can successfully approximate the solution. However, for initial value problems

---

*Equal contribution, order chosen by random coin flip. {maf820, ap6604}@nyu.edu

(IVPs), treating time as merely another spatial dimension results in complications for the neural network like catastrophic forgetting. Some heuristics have been developed to ameliorate this latter problem, such as increasing the collocation points as time progresses, but then the computational cost of training the neural network becomes impractical.

Recently, Du & Zaki (2021) and Bruna et al. (2022) have provided two methods that follow a novel *local-in-time* approach for training neural networks to solve IVPs by updating the network parameters sequentially through time rather than by having some fixed set of parameters to model the whole spatio-temporal domain. These methods have proven successful for a variety of PDEs, but they currently suffer from two shortcomings. First, the conditioning of the linear systems required to follow the ODE on the network parameters degrades over time, leading to longer solving times and ultimately to a complete breakdown of the solution. Second, the current methodologies lack the capacity to represent difficult initial conditions and solutions as their runtime scales cubically in the number of network parameters, limiting their ability to use large neural networks. In this work we provide a *local-in-time* IVP solver (*Neural-IVP*) that circumvents the shortcomings of Du & Zaki (2021) and Bruna et al. (2022) and thus enable us to solve challenging PDEs. In particular:

- Leveraging fast matrix vector multiplies and preconditioned conjugate gradients, we develop an approach that scales only linearly in the number of parameters, allowing us to use considerably larger neural networks and more data.
- We further improve the representational power and quality of the fit to initial conditions through the use of last layer linear solves and sinusoidal embeddings.
- We show how following the parameter ODE leads the network parameters to an increasingly poorly conditioned region of the parameter space, and we show how this relates to exact and approximate parameter symmetries in the network.
- Using regularization, restarts, and last layer finetuning, we are able to prevent the parameters from reaching these poorly conditioned regions, thereby stabilizing the method.

We provide a code implementation at `https://github.com/mfinzi/neural-ivp`.

## 2 BACKGROUND

Given a spatial domain $\mathcal{X} \subseteq \mathbb{R}^D$, we will consider the evolution of a time-dependent function $u : \mathcal{X} \times [0, T] \to \mathbb{R}^k$ which at all times belongs to some functional space $\mathcal{U}$ and with dynamics governed by

$$
\begin{aligned}
\partial_t u\,(x, t) &= \mathcal{L}\,[u]\,(x, t) && \text{for } (x, t) \in \mathcal{X} \times [0, T] \\
u\,(x, 0) &= u_0\,(x) && \text{for } x \in \mathcal{X} \\
u\,(x, t) &= h\,(x, t) && \text{for } x \in \partial\mathcal{X} \times [0, T]
\end{aligned}
$$

where $u_0 \in \mathcal{U}$ is the initial condition, $h$ is spatial boundary condition, and $\mathcal{L}$ is the (possibly non-linear) operator containing spatial derivatives. We can represent PDEs with higher order derivatives in time, such as the wave equation $\partial_t^2 \phi = \Delta \phi$, by reducing them to a system of first order in time equations $u := [\phi, \partial_t \phi]$, where in this example $\mathcal{L}[u_0, u_1] = [u_1, \Delta u_0]$.

**Global Collocation Methods**    The first approaches for solving PDEs via neural networks are based on the idea of sampling uniformly on the whole spatio-temporal domain and ensuring that the neural network obeys the PDE by minimizing the PDE residual (or a proxy for it). This approach was initially proposed by Dissanayake & Phan-Thien (1994) and Lagaris et al. (1998), which used neural networks as approximate solutions. However, recent advances in automatic differentiation, compute, and neural network architecture have enabled successful applications such as the Deep Galerkin Method (Sirignano & Spiliopoulos, 2018), Deep Ritz Method (Yu et al., 2018), and PINN (Raissi et al., 2019), which have revitalized interest in using neural networks to solve PDEs.

**Learning From Simulations**    Not all approaches use neural networks as a basis function to represent the PDE solution. Some approaches focus on directly learning the PDE operator as in Lu et al. (2019) or Kovachki et al. (2021), where the operator can be learned from simulation. However, as these methods typically use grids, their purpose is to accelerate existing solvers rather than tackling new problems. Other approaches that do not rely on collocation points exploit specific information of elliptic and semi-linear parabolic PDEs, such as E. et al. (2017) and Han et al. (2018).

## 2.1 GLOBAL PDE RESIDUAL MINIMIZATION

The most straightforward method for producing a neural network solution to initial values PDEs is similar to the approach used for boundary value problems of treating the temporal dimensions as if they were spatial dimensions and parameterizing the solution simultaneously for all times $u(x, t) = N_\theta(x, t)$. The initial and boundary conditions can be enforced through appropriate parameterization of the network architecture (Berg & Nyström, 2018), whereas the PDE is enforced through minimization of the training objective:

$$S(\theta) = \int_{\mathcal{X} \times [0,T]} r_\theta(x, t)^2 d\mu(x) dt = \int_{\mathcal{X} \times [0,T]} [(\partial_t u_\theta(x, t) - \mathcal{L}[u_\theta](x, t))^2] d\mu(x) dt$$

where the expectation is estimated via Monte Carlo samples over a chosen distribution of $\mu$ and times $t$.

Initial value PDEs have a local temporal structure where only values on the previous spatial slice are necessary to compute the next; however, global minimization ignores this property. Moreover, as the weights of the neural network must be used to represent the solution simultaneously at all times, then we must ensure that the neural network approximation does not forget the PDE solution learnt at earlier times (catastrophic forgetting). While Sirignano & Spiliopoulos (2018) and Sitzmann et al. (2020) take this approach, the downsides of avoiding catastrophic forgetting is to increase the computation spent by ensuring the presence of data from previous times.

## 2.2 LOCAL-IN-TIME METHODS

To circumvent the inherent inefficiency of the global methods, Du & Zaki (2021) and Bruna et al. (2022) propose a local-in-time method whereby the minimization problem gets converted into an ODE that the parameters satisfy at each point in time. In this approach, the PDE solution is given by $u(x, t) = N(x, \theta(t))$ for a neural network N, where the time dependence comes from the parameter vector $\theta(t)$ rather than as an input to the network. Thus, the network only represents the solution at a single time, rather than simultaneously at all times, and subsequently $\theta(t)$ can be recorded and no representational power or computational cost is incurred from preserving previous solutions. Assuming the PDE is one-dimensional, the PDE residual at a single time can be written as,

$$L(\dot{\theta}, t) = \int_{\mathcal{X}} r(x, t)^2 d\mu(x) = \int_{\mathcal{X}} (\dot{\theta}^\top \nabla_\theta N(x, \theta(t)) - \mathcal{L}[N](x, \theta(t)))^2 d\mu(x), \qquad (1)$$

since the time derivative is $\partial_t u(x, t) = \dot{\theta}^\top \nabla_\theta N(x, \theta)$.

Choosing the dynamics $\dot{\theta}$ of the parameters to minimize the instantaneous PDE residual error $L(\dot{\theta}, t)$ yields the (implicitly defined) differential equation

$$M(\theta)\dot{\theta} = F(\theta) \quad \text{and} \quad \theta_0 = \arg\min_\theta \int_{\mathcal{X}} (N(x, \theta) - u(x, 0))^2 d\mu(x), \qquad (2)$$

where $M(\theta) = \int_{\mathcal{X}} \nabla_\theta N(x, \theta) \nabla_\theta N(x, \theta)^\top d\mu(x)$ and $F(\theta) = \int_{\mathcal{X}} \nabla_\theta N(x, \theta) \mathcal{L}[N](x, \theta) d\mu(x)$. Once we find $\theta_0$ to fit the initial conditions, we have a fully specified system of differential equations, where we can advance the parameters (and therefore the solution $u(x, \theta(t))$) forward in time.

Since both $M(\theta)$ and $F(\theta)$ involve integrals over space, we can estimate them with $n$ Monte Carlo samples, yielding $\hat{M}(\theta)$ and $\hat{F}(\theta)$. We then proceed to solve the linear system $\hat{M}(\theta)\dot{\theta} = \hat{F}(\theta)$ at each timestep for the dynamics $\dot{\theta}$ and feed that into an ODE integrator such as RK45 (Dormand & Prince, 1980). For systems of PDEs such as the Navier-Stokes equations, the method can be extended in a straightforward manner by replacing the outer product of gradients with the Jacobians of the multi-output network $N$: $M(\theta) = \int_{\mathcal{X}} D_\theta N(x, \theta)^\top D_\theta N(x, \theta) d\mu(x)$ and likewise for $F$, which results from minimizing the norm of the PDE residual $\int_{\mathcal{X}} \|r(x, t)\|^2 d\mu(x)$.

Introducing some additional notation, we can write the Monte Carlo estimates $\hat{M}$ and $\hat{F}$ in a more illuminating way. Defining the Jacobian matrix of the network for different input points $J_{ik} = \frac{\partial}{\partial \theta_k} N(x_i, \theta)$, and defining $f$ as $f_i = \mathcal{L}[N](x_i, \theta)$, the PDE residual estimated via the $n$-sample

Monte Carlo estimator is just the least squares objective $\hat{L}(\dot{\theta}, t) = \frac{1}{n}\|J\dot{\theta} - f\|^2$. The matrices $\hat{M}(\theta) = \frac{1}{n}J^\top J$ and $\hat{F}(\theta) = \frac{1}{n}J^\top f$ reveal that the ODE dynamics is just the familiar least squares solution $\dot{\theta} = J^\dagger f = (J^\top J)^{-1}J^\top f$.

## 3    DIAGNOSING LOCAL-IN-TIME NEURAL PDE SOLVERS

The success of *local-in-time* methods hinges on making the PDE residual $L(\dot{\theta}, t)$ close to 0 as we follow the dynamics of $\dot{\theta} = \hat{M}(\theta)^{-1}\hat{F}(\theta)$. The lower the local error, the lower the global PDE residual $S(\theta) = \int L(\dot{\theta}, t)dt$ and the more faithfully the PDE is satisfied.

Even though $\dot{\theta}$ directly minimizes $L(\dot{\theta}, t)$, the PDE residual is not necessarily small and instead the value of $r(x, t)$ depends nontrivially on the network architecture and the values of the parameters themselves. While *local-in-time* methods have been applied successfully in several cases, there are harder problems where they fail unexpectedly. For example, they fail with unacceptably large errors in second-order PDEs or problems with complex initial conditions. In the following section, we identify the reasons for these failures.

### 3.1    REPRESENTATIONAL POWER

The simplest reason why *local-in-time* methods fail is because the neural networks do not have enough representational power. Having enough degrees of freedom and inductive biases in the network matters for being able to find a $\dot{\theta}$ in the span of $J$ which can match the spatial derivatives. The spatial derivatives $\mathcal{L}[N](x, \theta)$ of the PDE must be able to be expressed (or nearly expressed) as a linear combination of the derivative with respect to each parameter: $\frac{\partial}{\partial \theta_k}N(x, \theta)$, which is a different task than a neural network is typically designed for. The easiest intervention is to simply increase the number of parameters $p$, yielding additional degrees of freedom.

Increasing the number of parameters also improves the ability of the network to reconstruct the initial conditions, which can have knock-on effects with the evolution later in time. However, increasing the number of parameters and evolving through time following Du & Zaki (2021) and Bruna et al. (2022) quickly leads to intractable computations. The linear solves used to define the ODE dynamics require time $O(p^3 + p^2n)$ and use $O(p^2 + pn)$ memory, where $p$ represents the neural network parameters and $n$ is the number of Monte Carlo samples used to estimate the linear system. Therefore the networks with more than around $p = 5,000$ parameters cannot be used. Neural networks of these sizes are extremely small compared to modern networks which often have millions or even billions of parameters. In section 4.1, we show how our Neural-IVP method resolves this limitation, allowing us to use large neural networks with many parameters.

### 3.2    STABILITY AND CONDITIONING

In addition to lacking sufficient representational power, there are more subtle reasons why the *local-in-time* methods fail.

**Even when the solution is exactly representable, a continuous path $\theta^*(t)$ between the solutions may not exist.** Even if a network is able to faithfully express the solution at a given time $u(x, t) = N(x, \theta^*)$ for some value of $\theta^*$ in the parameter space, there may not exist a continuous path between these $\theta^*$ for different times. This fact is related to the implicit function theorem. With the multi-output function $H_i(\theta) = N(x_i, \theta) - u(x_i, t)$, even if we wish to satisfy $H_i = 0$ only at a finite collection of points $x_i$, the existence of a continuous path $\theta^*(t) = g(t)$ in general requires that the Jacobian matrix $D_\theta H = J$ is invertible. Unfortunately the Jacobian is not invertible because there exist singular directions and nearly singular directions in the parameter space, as we now argue.

**There exist singular directions of $J$ and $M$ as a result of symmetries in the network.** Each continuous symmetry of the network will produce a right singular vector of $J$, regardless of how many points $n$ are used in the Monte Carlo estimate. Here we define a continuous symmetry as a parameterized transformation of the parameters $T_\alpha : \mathbb{R}^p \to \mathbb{R}^p$ defined for $\alpha \in (-\epsilon, \epsilon)$, in a neighborhood of the identity $T_0 = \mathrm{Id}$, and $T_\alpha$ has a nonzero derivative with respect to $\alpha$ at the identity. For convenience, consider reparametrizing $\alpha$ to be *unit speed* so that $\|\partial_\alpha T_\alpha(\theta)\| = 1$.

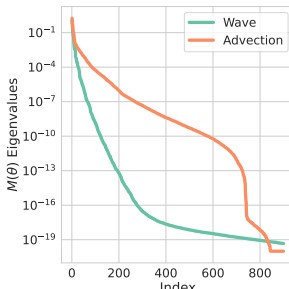 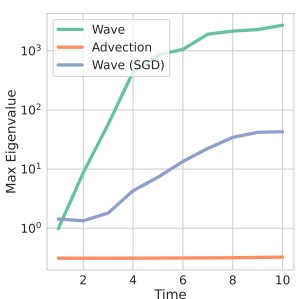 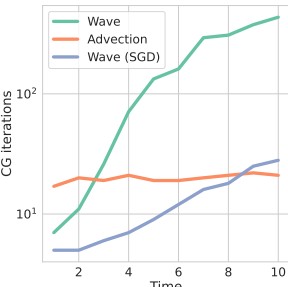

Figure 1: The conditioning of the linear systems needed to solve the ODE on the network parameters increases for challenging PDEs like the wave equation, but not for others like the advection equation. (Left): Eigenvalue spectrum of $M(\theta)$ matrix at initialization. (Middle): Growth of largest eigenvalue of $M(\theta)$ over time. (Right): Number of preconditioned CG iterations required to solve the linear system to a specified tolerance of $\epsilon = 10^{-7}$.

**Theorem 1.** *Suppose the network* $\mathrm{N}(x, \theta)$ *has a continuous parameter symmetry* $T_\alpha$ *which preserves the outputs of the function:* $\forall \theta, x : \mathrm{N}(x, T_\alpha(\theta)) = \mathrm{N}(x, \theta)$, *then*

$$v(\theta) = \partial_\alpha T_\alpha(\theta)\big|_{\alpha=0} \tag{3}$$

*is a singular vector of both* $J$ *and* $M$.

**Proof:** *Taking the derivative with respect to* $\alpha$ *at* $0$, *from the chain rule we have:* $0 = \partial_\alpha\big|_0 N(x, T_\alpha(\theta)) = \nabla_\theta N(x, \theta)^\top \partial_\alpha T_\alpha(\theta)\big|_{\alpha=0}$. *As this expression holds for all* $x$, $J(\theta)v(\theta) = 0$ *and* $M(\theta)v(\theta) = 0$.

As Dinh et al. (2017) demonstrated, multilayer perceptrons using ReLU nonlinearities have a high-dimensional group of exact parameter symmetries corresponding to a rescaling of weights in alternate layers. Furthermore even replacing ReLUs with alternate activation functions such as Swish (Ramachandran et al., 2017) does not solve the problem, as these will have approximate symmetries which will produce highly ill-conditioned $M$ and $J$ matrices.

**Theorem 2.** *An approximate symmetry* $\forall x : \|N(x, T_\alpha(\theta)) - N(x, \theta)\|^2 \leq \epsilon \alpha^2$ *will produce nearly singular vectors* $v(\theta) = \partial_\alpha T_\alpha(\theta)\big|_{\alpha=0}$ *for which*

$$v^\top M v < \epsilon, \tag{4}$$

*and therefore the smallest eigenvalue of* $M$ *is less than* $\epsilon$.

**Proof:** *See Appendix A*

Additionally, the rank of the Monte Carlo estimate $\hat{M} = \frac{1}{n} J^\top J$ using $n$ samples is at most $n$, and there is a $p - n$ dimensional manifold of parameters which match the function values at the sample points $\forall i = 1, ..., n : \mathrm{N}(x_i, \theta) = \mathrm{N}_i$ rather than over the whole domain. In Figure 1 (left), we show empirically that the eigenspectrum of $\hat{M}$ is indeed deficient and highly ill-conditioned, with a long tail of small eigenvalues.

Hence some form of regularization such as $[M(\theta) + \mu I]\dot{\theta}$ is necessary to have a bounded condition number, $\kappa(M(\theta) + \mu I) = (\lambda_1 + \mu)/(0 + \mu)$.

Furthermore, as seen in Figure 1 (middle) the conditioning of the linear system (equation 2) deteriorates over time. This deterioration worsens in more challenging PDEs like second-order wave equation in contrast to the easier advection equation. Even when using a dense solver, the quality of the solves will degrade over time, leading to increased error in the solution. When using an iterative method like CG shown in Figure 1 (right), the runtime of the method will increase during the evolution and eventually not be able to meet the desired error tolerance. In contrast, if we instead take snapshots of the solution at different times and fit it directly with SGD, we find that the conditioning is much better as shown by the green curve in Figure 1.

Making sense of this observation, we can learn from the ways that neural networks are typically used: in conjunction with stochastic gradient descent. In general when training neural networks with

SGD, we must chose carefully the initialization to make the problem well-conditioned (Mishkin & Matas, 2015). Many initializations, such as setting the scale of the parameters with drastically different values between layers, will lead either to diverging solutions or no progress on the objective. With the right initialization and a good choice of hyperparameters, the optimization trajectory will stay in a well-conditioned region of the parameter space. However, many bad regions of the parameter space exist, and a number of architectural improvements in deep learning such as batch normalization (Ioffe & Szegedy, 2015) and skip connections (He et al., 2016) were designed with the express purpose of improving the conditioning of optimization while leaving the expressive power unchanged. Unfortunately, while SGD optimizers stay in a well-conditioned regions of the parameter space, equation (2) does not.

To see this, consider a singular vector $v$ of $J$ which has a very small singular value $\sigma_v$ (due to an approximate symmetry or otherwise). Analyzing the solution, we see that the projection of the parameters along the singular vector evolves like: $v^\top \dot{\theta} = v^\top J^\dagger f = \sigma_v^{-1} u_v^\top f$, where $Jv = \sigma_v u_v$. A small singular value leads to a large change in that subspace according to the evolution of the ODE. Considering the approximate rescaling symmetry for swish networks, this points the dynamics directly into amplifying the difference between size of weights in neighboring layers and worsening the conditioning, precisely what was identified as *sharp minima* from Dinh et al. (2017). We describe how this problem can be circumvented by our method as described in section 4.2.

# 4    NEURAL IVP

Drawing on these observations, we introduce Neural IVP, a method for solving initial value PDEs that resolves the scalability and numerical stability issues limiting current *local-in-time* methods. We also enhance the Neural IVP, through projections to different regions of the parameter space, fine-tuning procedures, and mechanisms to increase the representation power of the neural networks.

## 4.1    IMPROVING REPRESENTATIONAL POWER AND SCALABILITY

To evaluate the representational power of the network, we examine fitting a complex initial condition typically present in the later evolution of an entangled system, and which will contain components at many different frequencies. We fit a sum of two Gaussians of different sizes modulated by frequencies pointing in different directions in 3-dimensions within the cube $[-1, 1]^3$, with the slice through $z = 0$ shown in Figure 2 (left). The function is defined as the sum of two Gaussian like wave packets, modulated by spatial frequencies pointing in different directions:
$u_0 (x) = 30 \left(2\pi s_1^2\right)^{-1} e^{-||v||_2^2/(2s_1^2)} \cos\left(2\pi f x^\top \hat{n}\right) + 24 \left(2\pi s_2^2\right)^{-1} e^{-||w||_2^2/(2s_2^2)} \cos\left(2\pi f x^\top \hat{m}\right)$.
where $v = x - 0.5(\hat{x}_2 + \hat{x}_3)$, $w = x + (\hat{x}_1 + \hat{x}_2 + \hat{x}_3)/6$ also $\hat{n} = (\hat{x}_1 + \hat{x}_2)/\sqrt{2}$ and $\hat{m} = 2^{-1}(\hat{x}_2 + x_1\hat{x}_1/3)$.

By changing the frequency parameter $f$, we can investigate how well the network is able to fit fine-scale details. We introduce three substantial improvements to the models over those used in Evolutional Deep Neural Networks (EDNN) of Du & Zaki (2021) in order to improve their representational power, and we evaluate the impact of these changes in Figure 2 (middle), starting with the exact 4-layer 30 hidden unit tanh nonlinearity MLP architecture used in EDNN.

**Increasing Number of Parameters and Scalability** As shown in Figure 2 (right), the number of network parameters has a large impact on its representational power, not just for solving the initial conditions but also for finding a $\dot{\theta}$ that achieves a low PDE residual. Increasing the number of parameters substantially reduces the approximation error, especially at high frequencies which are challenging for the model. While dense solves prohibit scaling past 5,000 parameters, we show that our solves can be performed much faster making use of the structure of $\hat{M}$. Matrix vector multiplies with the matrix $\hat{M}(\theta) = \frac{1}{n}J^\top J$ can be implemented much more efficiently than using the dense matrix. Making use of Jacobian-vector-products implemented using automatic differentiation (such as in JAX (Bradbury et al., 2018)), we can implement a matrix vector multiply using 2 Jacobian-vector-products:
$$\hat{M}(\theta) v = \frac{1}{2n}\nabla_v \|Jv\|^2, \tag{5}$$
which takes $O(n + p)$ time and memory for a single matrix-vector product, sidestepping memory bottlenecks that are usually the limiting factor. Then, with these efficient matrix-vector products,

Figure 2: (Left): Example wave packet initial condition with varying levels of fine details parameterized by the frequency $f$. (Middle): Impact of Neural-IVP improvements in the model on the initial condition fit relative error across different levels of difficulty of the solution as parameterized by the frequency $f$, yielding an improvement of $1-2$ orders of magnitude. (Right): Initial condition fit with all Neural-IVP interventions but with varying number of parameters in the model (shown by the colors). Note that the largest networks which can be used by the dense method are only 5000 parameters.

we can use a scalable Krylov subspace routine of conjugate gradients (CG). To further accelerate the method, we construct a Nyström preconditioner (Frangella et al., 2021), drastically reducing the number of CG iterations. Using this approach, each solve takes time $O((n + p)\sqrt{\kappa})$ where $\kappa(P^{-1}\hat{M})$ is the condition number of the preconditioned matrix $\hat{M}$, rather than $O(p^3 + p^2 n)$ time of dense solves. These improvements to the runtime using the structure of $\hat{M}$ mirror the sparse and Kronecker structures used by finite difference methods.

**Sinusoidal Embedding**   We make several architectural enhancements to the networks used in Du & Zaki (2021) and Bruna et al. (2022), which improve the quality of the initial condition fit and the error when evolving forward in time. Notably, we find that using the sinusoidal embedding (Mildenhall et al., 2021) substantially improves the ability of the network to represent higher frequency details in the solution. In contrast to the original form, we use the featurization

$$\gamma(x) = [\sin(2^k x \tfrac{\pi}{2})2^{-\alpha k}]_{k=0}^L + [\cos(2^k x \tfrac{\pi}{2})2^{-\alpha k}]_{k=0}^L, \tag{6}$$

which scales the magnitude of the high frequency (large $\omega$) components down by $1/\omega^\alpha$. While $\alpha = 0$ (the original sinusoidal embedding) works the best for fitting an initial signal (the only requirement needed for Neural Radiance Fields (Mildenhall et al., 2021)), the derivatives of $\gamma$ will not be well behaved as the magnitude of the largest components of $\gamma'(x)$ will scale like $2^L$ and $\gamma''(x)$ will scale like $2^{2L}$. We find setting $\alpha = 1$ to be the most effective for both first order and 2nd order PDEs. Figure 2 (middle) shows the sinusoidal embedding helps the model represent complex functions.

**Last Layer Linear Solves**   To further improve the quality of the initial condition fit, after training the network on the initial condition, we recast the fitting of the last layer of the network as solving a linear least squares problem. The network up until the last layer can be considered as features $N(x) = w^\top \phi_\theta(x) + b$ over a fixed set of collocation points $X$. We can then solve the minimization problem with respect to the final layer weights $w, b$

$$\min_{w,b} \|w^\top \phi_\theta(X) + b - u_0(X)\|^2, \tag{7}$$

which can be solved to a higher level of precision and achieve a lower error when compared to the values of the last layer obtained without tuning from the full nonlinear and stochastic problem.

Combining these three improvements of scalability, sinusoidal embeddings, and last layer linear solves (head tuning), we are able to reduce the representation error of the networks by 1-2 orders of magnitude across different difficulties of this challenging 3 dimensional problem.

## 4.2 STABILITY AND CONDITIONING

**Preconditioning**   In section 3.2 we discussed how even for easier PDEs, the symmetries in the neural networks generate badly conditioned linear systems for the ODE on the parameters. To counteract this negative effect on our CG solver, we use the highly effective and scalable randomized Nyström preconditioner (Frangella et al., 2021). As discussed in Frangella et al. (2021),

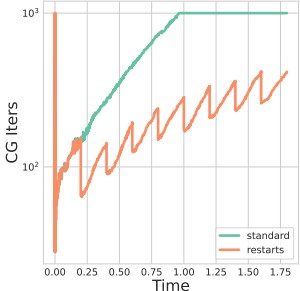 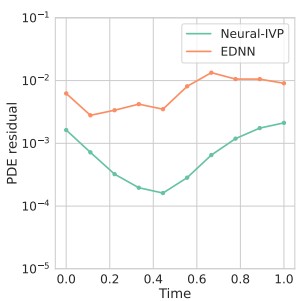 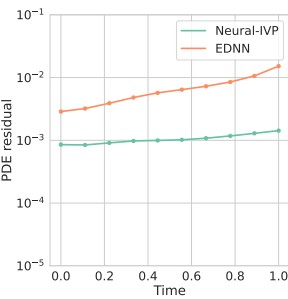

Figure 3: (Left): Restarts improve the condition of the linear systems. Here we cap the number of CG iterations at 1000, but without restarts the number required to reach a desired error tolerance will only continue to increase. Neural-IVP achieves an order of magnitude better PDE residual than EDNN on the Fokker-Plank equation (middle) and on the Vlasov equation (right).

this preconditioner is close to the optimal truncated SVD preconditioner and it is empirically impressive. To use this preconditioner, we first construct a Nyström approximation of $M(\theta)$: $M_{\text{nys}}(\theta) = (M(\theta)\Omega)(\Omega^\top M(\theta)\Omega)^{-1}(M(\theta)\Omega)^\top$ using a Gaussian random subspace projection $\Omega \in \mathbb{R}^{p \times \ell}$, where $\ell$ denotes the subspace rank. Then, using the SVD of the approximation $M_{\text{nys}}(\theta) = U\hat{\Lambda}U^\top$ we can construct a preconditioner (where $\nu$ is a small regularization) as follows:

$$P = \tfrac{1}{\hat{\lambda}_\ell + \nu} U(\hat{\Lambda} + \nu I)U^\top + (I - UU^\top).$$

This preconditioner closely approximates the optimal truncated SVD preconditioner when the eigenspectrum $\hat{\Lambda}$ resembles that of the original problem. The cost of using this preconditioner are $\ell$ matrix-vector-multiplies (MVMs) and a Cholesky decomposition of $O(\ell^3)$, where in our problems $\ell \in \{100, 200, 300\}$.

**Projection to SGD-like regions** Following the ODE on the parameters leads to linear systems whose condition worsens over time as seen on the middle panel of Figure 1. In that plot it is also visible how the conditioning of the systems is much lower and does not increase as rapidly when the neural network is trained to fit the PDE solution using SGD. In principle we would opt for the SGD behavior but we do not have access to fitting the ground truth PDE solution directly. Instead, when the condition number grows too large, we can refit against our neural predictions using SGD and thus start from another location in the parameter space. That is, every so often, we solve $\theta^{\text{SGD}}(t) = \arg\min_\theta \int_{\mathcal{X}} (\text{N}(\theta, x) - \text{N}(\theta(t), x))^2 \, d\mu(x)$ where $\theta(t)$ is our current parameters at time $t$. Performing restarts in this way considerably improves the conditioning. As seen in Figure 3 (left) the number of CG iterations increases substantially slower when using the SGD restarts.

### 4.3 VALIDATING OUR METHOD

We validate our method with three different PDEs: the wave equation (3+1), the Vlasov equation (6+1) and the Fokker-Planck equation (8+1). For more details on these equations, see Appendix B. For the wave equation we make comparisons against its analytic solution, while for the remaining equations we compare using the PDE residual (evaluated on different samples from training), additional details about the experiment setup can be found in appendix B. For the wave equation, Neural-IVP achieves the lowest error beating EDNN and finite differences evaluated on a $100 \times 100 \times 100$ grid as seen in Figure 6. For the remaining equations Neural-IVP achieves an order of magnitude lower PDE residual compared to EDNN[1] as seen in the middle and right panels of Figure 3.

---

[1]While we would like to benefit from the improved sampling distribution proposed by Bruna et al. (2022), unfortunately since the solution is not a probability distribution there is no clear measure to sample from.

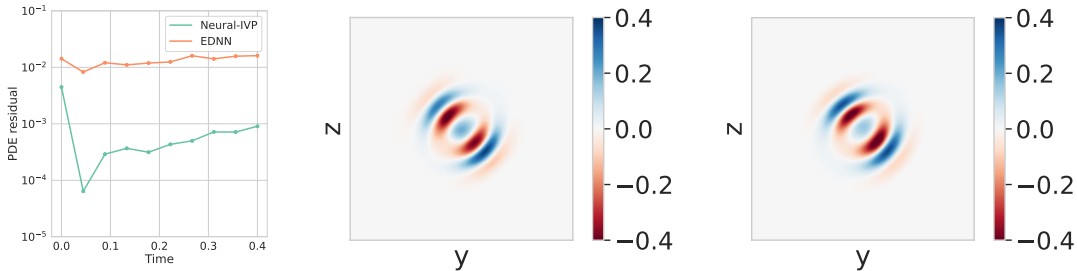

Figure 4: (Left): Neural-IVP's PDE residual over time for wave maps compared to EDNN. (Middle): Neural-IVP solution for the wave maps at $t = 0.36$ at a $x = 0$ slice. (Right): Finite difference solution for the wave maps at the same slice.

## 5 SOLVING CHALLENGING HYPERBOLIC PDEs

We now turn to a challenging PDE: the wave maps equation. This equation is a second-order hyperbolic PDE that often arises in general relativity and that describes the evolution of a scalar field in a curved (3+1) dimensional spacetime. Following Einstein's tensor notation from differential geometry, the wave maps equation can be expressed as

$$g^{\mu\nu}\nabla_\mu\nabla_\nu\phi = g^{\mu\nu}\partial_\mu\partial_\nu\phi - g^{\mu\nu}\Gamma^\sigma_{\mu\nu}\partial_\sigma\phi = 0, \tag{8}$$

where $g$ is the metric tensor expressing the curvature of the space, $\Gamma^\sigma_{\mu\nu}$ are the Christoffel symbols, combinations of the derivatives of the metric, and $\partial_\mu$ are derivatives with respect to components of both space and time. For the metric, we choose the metric from a Schwarzschild black hole located at coordinate value $c = -2\hat{x}$ in a Cartesian-like coordinate system:

$$g_{\mu\nu}dx^\mu dx^\nu = -(1 - r_s/r_c)dt^2 + [\delta_{ij} + \frac{1}{r_c^2(r_c/r_s-1)}(x_i - c_i)(x_j - c_j)]dx^i dx^j,$$

where $r_c = \|x - c\| = \sqrt{\sum_i (x_i - c_i)^2}$ and $r_s = 2M$ is the radius of the event horizon of the black hole, and we choose the mass $M = 1/2$ so that $r_s = 1$. We choose a wave packet initial condition and evolve the solution for time $T = .5 = 1M$ inside the box $[-1, 1]^3$, with the artificial Dirichlet boundary conditions on the boundary $\partial[-1, 1]^3$. Here the event horizon of the black hole lies just outside the computational domain and boundary conditions meaning that we need not worry about complications on and inside the horizon, and instead the scalar field only feels the effect of the gravity and is integrated for a time short enough that it is not yet pulled inside.

While we do not have an analytic solution to compare to, we plot the relative error of the PDE residual averaged over the spatial domain in Figure 4 which is consistently small. We also compare the solution at the time $T = 0.36$ of our solver solution against the finite difference solution run at a spatial grid size of $150 \times 150 \times 150$ which is the largest we were able to run with our optimized sparse finite difference solver before running out of memory. Despite the challenging nature of the problem, Neural-IVP is able to produce a consistent solution for this task. Finally, we present an ablation study in Figure 6 showing the gains of using the sinusoidal embedding and of scaling the neural network size and grid for this experiment.

## 6 DISCUSSION

There are many PDEs of interest that are massively complex to simulate using classical methods due to the scalability limitations of using grids and meshes. At the same time, neural networks have shown promise for solving boundary value problems but the current methods for solving initial value problems can be unstable, deficient in scale and in representation power. To ameliorate these deficiencies, we presented Neural-IVP, a *local-in-time* method for approximating the solution to initial value PDEs. Neural-IVP is a compelling option for problems that are computationally challenging for classical methods like the (3+1) dimensional wave maps equation in section 5. Continuous effort on this front will empower researchers and engineers to simulate physical PDEs which lie at the boundary of what is currently possible to solve, allowing prototyping and experimentation without the massive complexity of modern large scale grid-based solvers involving mesh generation, mesh refinement, boundaries, excision, parallelization, and communication.

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

## A   APPROXIMATE SYMMETRIES YIELD SMALL EIGENVALUES

Suppose that the network $\mathrm{N}$ has an approximate symmetry in the parameters, meaning that there exists a value $\epsilon$ for which

$$\forall x, \theta : \|\mathrm{N}(x, T_\alpha(\theta)) - \mathrm{N}(x, \theta)\|^2 \le \epsilon \alpha^2. \tag{9}$$

holds for $\alpha$ in a neighborhood of $0$. If this is the case, we can rearrange the inequality and take the limit as $\alpha \to 0$:

$$\lim_{\alpha \to 0} \|\mathrm{N}(x, T_\alpha(\theta)) - \mathrm{N}(x, \theta)\|^2 / \alpha^2 \le \epsilon. \tag{10}$$

As the limit $\lim_{\alpha \to 0} \frac{\mathrm{N}(x, T_\alpha(\theta)) - \mathrm{N}(x, \theta)}{\alpha} = \frac{\partial}{\partial \alpha} \mathrm{N}(x, T_\alpha(\theta))$ exists, we can interchange the limit and norm to get

$$\|\nabla_\theta \mathrm{N}^\top v\|^2 = v \nabla_\theta \mathrm{N} \nabla_\theta \mathrm{N}^\top v \le \epsilon, \tag{11}$$

since $\frac{\partial}{\partial \alpha} \mathrm{N}(x, T_\alpha(\theta)) = \nabla_\theta \mathrm{N}^\top v$ for $v(\theta) = \partial_\alpha T_\alpha(\theta)\big|_{\alpha=0}$. Recalling that $M(\theta) = \int \nabla_\theta \mathrm{N} \nabla_\theta \mathrm{N}^\top d\mu(x)$, we can take the expectation of both sides of the inequality with respect to $\mu$, producing

$$v^\top M v < \epsilon. \tag{12}$$

## B   EXTENDED EXPERIMENTAL RESULTS

In this section we expose additional experimental details that were not fully covered in section 4.3.

### B.1   WAVE EQUATION

For this experiment we use the 3 dimensional wave equation $\partial_t^2 u = \Delta u$, that has a few well known analytic solutions. Even this equation is computationally taxing for finite difference and finite element methods. We use the radially symmetric outgoing wave solution $u(x, t) = f(t - \|x\|)/\|x\|$ with $f(s) = 2s^2 e^{-200s^2}$ and integrate the initial condition forward in time by $T = .5$ seconds. In Figure 6 we compare to the analytic solution each of the solutions produced by Neural-IVP, EDNN (Du & Zaki, 2021), and finite differences evaluated on a $100 \times 100 \times 100$ grid with RK45 set to a $10^{-4}$ tolerance. Despite the fact that this initial condition does not contain fine scale detail that Neural-IVP excels at, Neural-IVP performs the best among the three solvers, and faithfully reproduces the solution as shown in Figure 6 (left).

### B.2 VLASOV EQUATION

The Vlasov equation is a PDE that describes the evolution of the density of collisionless but charged particles in an electric field, expressed both as a function of position and velocity. This equation has spatial dimension 6 and takes the following form

$$\partial_t u\left(x, v, t\right) + v^\top \nabla_x u\left(x, v, t\right) + \frac{q}{m} E(x, t)^\top \nabla_v u\left(x, v, t\right) = 0$$

where the vector $x \in \mathbb{R}^3$ represents the position of the particles and $v \in \mathbb{R}^3$ represents the velocity, and $u$ represents a normalized probability density over $x, v$. Here $q$ is the charge of the particles and $m$ is their mass (both of which we set to 1). In a self contained treatment of the Vlasov equation, the electric field $E(x, t)$ is itself induced by the density of charged particles: $E(x, t) = -\nabla_x \phi(x, t)$ and the potential $\phi$ is the solution to the Poisson equation $\Delta \phi = -\rho$ where $\rho(x, t) = \int q u(x, v, t) dv$. However, to simplify the setting to a pure IVP, we assume that $E(x, t)$ is some known and fixed electric field.

For this particular example, we choose $E\left(x\right) = \nabla_x \exp(-\|x\|_2^2)$ and the initial condition is a product of two Gaussians

$$u_0\left(x, v\right) = \mathcal{N}(x; 0, .3^2 I)\mathcal{N}(v; 0, .3^2 I),$$

corresponding to the Maxwell-Boltzmann distribution over velocities and a standard Gaussian distribution over position, and we solve the problem on the cube $[-1, 1]^6$.

### B.3 FOKKER-PLANCK

For the Fokker-Planck equation, we choose the harmonic trap for a collection of $d = 8$ interacting particles from Bruna et al. (2022), giving rise to the equation

$$\partial_t u\left(x, t\right) = D\Delta u(x, t) - \nabla \cdot (hu),$$

where $h(x) = (a - x) + \alpha(\mathbf{1}\mathbf{1}^\top/d - I)x$. We choose $a = (0.2)\mathbf{1}$ along with constants $D = .01$ and $\alpha = 1/4$.

We solve this equation in $d = 8$ dimensions with the initial condition

$$u_0\left(x\right) = \left(\frac{3}{4}\right)^d \Pi_{i=1}^d (1 - x_i^2)$$

which is a normalized probability distribution on $[-1, 1]^d$. We use the same Dirichlet boundary conditions for this problem.

Additionally, since we can naturally increase the dimensionality of the Fokker-Planck equation, we explore up to what dimension Neural-IVP can give reasonable PDE residuals. As seen in Figure 5, Neural-IVP can still give solutions for $d = 20$. For dimensions higher that 20, the linear system solvers cannot converge to the desire tolerance needed to evolve the parameters. We warn that these higher dimensional solutions are not guaranteed to be of high quality since the PDE residual estimation also worsens as we increase the spatial dimension.

## C HYPERPARAMETERS

For our experiments we used the following hyperparameters for Neural-IVP in our experiments unless otherwise specified:

1. RK45 Integrator with rtol: 1e-4 (for all equations except wave maps, which uses RK23)
2. Number of Monte Carlo samples: 50K for wave maps and 10-20K for all other PDEs
3. Maximum CG iterations: 1,000
4. CG tolerance: 1e-8
5. Nyström preconditioner rank: 200-350
6. Linear system regularization: 1e-6
7. Initial fit iterations, optimizer and learning rate: 50K, ADAM, 1e-3

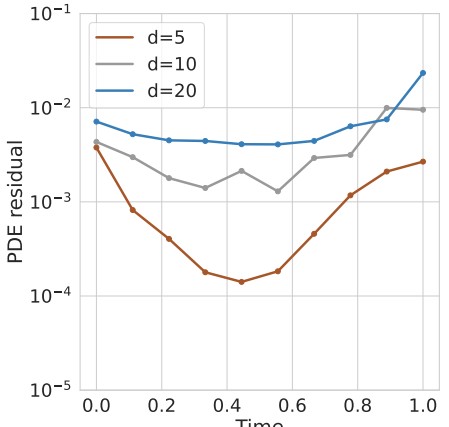 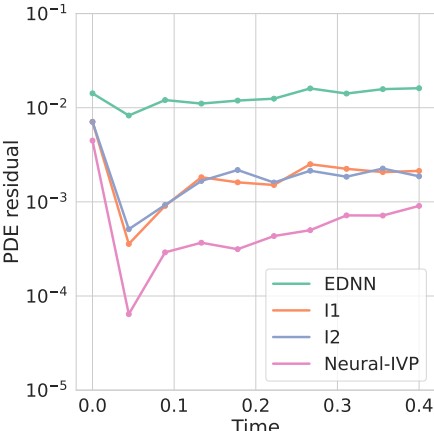

Figure 5: (Left): Neural-IVP is able to provide solutions up to dimension $20 + 1$ for the Fokker-Planck equation. Higher dimensions break-down the linear systems to evolve the PDE parameters. (Right): Interventions to transform EDNN to Neural-IVP. (I1) First, add a sinusoidal embedding before the MLP. (I2) Second, use head finetuning (in this case there is no notable improvement as the initial condition does not posses finer details). Finally, scale the neural network and the grid size. Note that this is only possible due to our scalable and efficient construction.

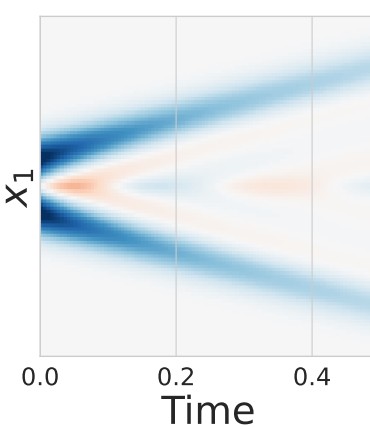 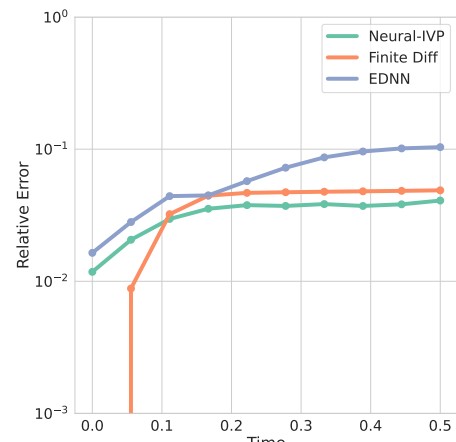

Figure 6: (Left): Neural-IVP fit of the wave equation through time. (Right): Neural-IVP performs slightly better than a finite difference method on the 3D wave equation.

8. Floating point precision: double

9. Number of restarts: 10

The neural network architecture we use is a simple MLP with 3 hidden layers, each with 100 hidden units, $L = 5$ for the highest frequency power in the sinusoidal embedding, and the network uses swish nonlinearities. The initial sinusoidal embedding values $\gamma(p)$ are scaled by 1.5 before feeding into the network.

---

**Algorithm 1** NEURAL-IVP

1: **Input:**
  1. **IVP**: Initial condition $u_0(x)$, PDE rhs operator $\mathcal{L}[u](x,t)$, integration time $T$
  2. **Design Choices**: neural network architecture $N(x,\theta)$, sampling distribution $\mu$
  3. **Hyperparameters**: $n$ number of Monte Carlo samples, `ODE_TOL`, `CG_TOL`, regularization $\mu$, preconditioner rank $r$

2: **Output:** Solution $u(x,t)$ at specified times $t_1, \ldots t_N \leq T$

3: **function** NEURAL-IVP
4:     $\theta \leftarrow \texttt{FitFunction}(u_0)$
5:     $\Delta t \leftarrow 20T \texttt{ODE\_TOL}$
6:     **while** $t < T$ **do**
7:         **if** Sufficient time since last restart **then**
8:             $\theta \leftarrow \texttt{FitFunction}(N(\cdot, \theta))$
9:         $\theta, \Delta t \leftarrow \texttt{Adaptive\_RK23\_Step}(\text{Dynamics}, \theta, \Delta t, \texttt{ODE\_TOL})$
10:         $t \leftarrow t + \Delta t$
          **return** $[N(\cdot, \theta_{t_1}), \ldots, N(\cdot, \theta_{t_N})]$

11: **function** FITFUNCTION(u)
12:     $\theta = \arg\min_\theta \mathbb{E}_{x \sim \mu} \|N(x, \theta) - u(x)\|^2$ minimized with Adam
13:     Separate last layer weights $W$ from $N(x, \theta) = W^\top \Phi_\theta(x)$ (including bias)
14:     Solve for $W$ from regularized least squares over samples $X$:
15:     $W \leftarrow (\Phi(X)^\top \Phi(X) + \lambda I)^{-1} \Phi(X)^\top u(X)$
16:     Assemble $\theta \leftarrow [\theta_{[:-1]}, W]$
          **return** $\theta$

17: **function** DYNAMICS($\theta$)
18:     Let the Jacobian vector product of $N(x_i, \theta)$ with $v$ taken with respect to $\theta$ be $DN(x_i, \theta)v$
19:     Construct efficient MVM $\hat{M}(\theta)v := \nabla_v \frac{1}{2n} \sum_{i=1}^n \|DN(x_i, \theta)v\|^2$
20:     Construct RHS $\hat{F}(\theta) = \nabla_v \frac{1}{2n} \sum_{i=1}^n \mathcal{L}[N](x_i, \theta)^\top DN(x_i, \theta)v$
21:     Construct rank-$r$ Nystrom predonditioner $P$ using $\hat{M}(\theta)$ MVMs
22:     Solve $(\hat{M}(\theta) + \mu I)\dot{\theta} = \hat{F}(\theta)$ for $\dot{\theta}$ using conjugate gradients with preconditioner $P$
23:     **return** $\dot{\theta}$

---

## D   NEURAL-IVP PSEUDO-CODE

The pseudo-code for Neural-IVP is present in Algorithm 1. For reduced clutter, have omitted the logic for setting step sizes so that $\theta$ will be output by the integrator at the specified times $t_1, t_2, \ldots t_N$. Sampling RNG state is updated outside the RK23 step but inside the time evolution while loop.

