# OpenReview forum: "A Stable and Scalable Method for Solving Initial Value PDEs with Neural Networks"
_ICLR.cc/2023/Conference — ICLR 2023 poster_

### Official Review · Reviewer_ZCbx · 2022-10-20

**Confidence:** 4
**Clarity, Quality, Novelty And Reproducibility:** The paper is clear.
**Correctness:** 2
**Technical Novelty And Significance:** 2
**Empirical Novelty And Significance:** Not applicable
**Recommendation:** 5

**Strength And Weaknesses:**

Strengths
- The scalability and numerical stability issues of local in time methods are analyzed.
- New techniques are proposed to address those issues.

Weaknesses
- A few techniques are used together to improve the local in time methods. Each of these techniques is not totally new. It is also unclear how important of each technique.
- There is only one example in this paper. Many more examples are needed.
- Even for this example, there are not sufficient convincing results. There are no quantitative errors.
- In the paper, it claims about “high dimensional” PDEs. However, the example in Section 5 is a 3D problem, not high dimensional at all. Examples of at least 10 dim are required.
- If time dependent 3D problems are “high dimensional”, then many existing papers have solved time dependent 3D problems by deep learning methods, such as physics-informed neural networks (PINNs). A lot of relevant literature review is missing.
- There are no comparisons with other methods (either deep learning or conventional methods) in terms of accuracy and speed.
- In Introduction, “Rather than compute costs which scale exponentially in the dimension, they instead typically have error …”. There are no definitions of N and epsilon. This sentence is not a rigorous mathematical statement. Also, references are required.
- In Introduction, “more that 3 dimensions”. A typo here.


**Summary Of The Paper:**

The paper developed Neural-IVP, an ODE based IVP solver, to solve initial value PDEs. Neural-IVP is based on the local in time methods, but the authors show that local in times methods have issue of representational power and numerical stability. Neural-IVP is developed by addressing these two issues.

**Summary Of The Review:**

The paper proposed extensions of local in time method, but the experiments are not convincing.

---

> ### Author Response · Authors · 2022-11-18
> **Author Response**
>
> Thank you for your review and feedback. **While we respect your viewpoint, it does not accurately reflect the novel contributions and evaluation that exists in the paper**. Each technique that we introduced has a clear benefit of addressing a current limitation that prevents the state-of-the-art methods of solving a hard PDE such as the second-order wave equation. We empirically evaluated the different interventions and improvements on representational power in figure 2, and we evaluated the positive impact the solution refits have on the condition number (and hence runtime and stability) in figure 3 left. We do directly compare both against conventional and alternative deep learning approaches for solving the PDEs: for the 3d wave equation which has an analytic solution we compare the relative errors in what is now fig 6 (previously fig 3 right) both against the classical finite difference solver and against a deep learning method (EDNN). For the wave maps equation, we do not have the analytic solution, so there is no relative error to compute, but we do compare the PDE residual and evaluate it against the EDNN method (which is a quantitative comparison). We have also now added more comparisons such as that of Neural-IVP against EDNN on the Vlasov and Fokker-Planck equations in the updated Figure 3.
>
> You assert that "Several of the paper’s claims are incorrect or not well-supported". Given this assertion, could you elaborate which claims in the paper are incorrect?
>
> In response to your more detailed questions,
>
> * **On number of benchmark PDEs and high dimensions**. We focused our experiments on two types of equations: a second-order wave equation (section 4.3) and on a wave maps equation (section 5) with the implicit understanding that our method would equally perform on easier problems. However, we agree that adding a broader set of equations would strengthen the papers and have done so in section 4.3, including the (6+1) dimensional Vlasov equation and (8+1) dimensional Fokker-Planck equation. These PDEs are already extremely high dimensional in the context of numerical solvers for PDEs, and are clear examples of PDEs in high dimensions (we also add an investigation of how our solution quality changes with increased dimension in figure 5 (right). You claim that “[t]here are no quantitative errors” but we plotted the PDE residual in Figure 4 which is a quantitative error, as seen in equation (1) in section 2.2. Finally, you claim “[e]ven for this example, there are not sufficient convincing results”. Our method is not only able to beat state-of-the-art solvers but also solve PDEs that are not achievable with previous methods. In our perspective this is a milestone.
>
> * **On novelty**. Our work introduces several novel methodological contributions. First, we translate the neural PDE problem as one that can be solved through Krylov subspace methods, where we define a fast MVM through JVPs as explained in section 4.1. Second, we propose the use of sinusoidal embeddings and last layer solves as explained in section 4.1. Third, we introduced a routine to project back to “SGD-like” regions in the parameter space as explained in section 4.2. Our method is the first neural PDE solver to employ all the previous techniques with clear theoretical and empirical benefits.
>
> * **Citations**. We cite all global collocation methods including PINNs. Moreover, we explain in the abstract, in the introduction, and section 2.1 how neural networks methods like PINNs have enjoyed success in applications of boundary value problems but they suffer from catastrophic forgetting on initial value problems. We are not the first to raise this issue. In Du & Zaki (2021) the whole introduction explains the limitations of PINNs as well as the introduction and related works section in Bruna et al. (2022).
> * We are familiar with the literature that uses neural networks to solve PDEs, especially since this is a nascent area. To the best of our knowledge we cited all the applicable papers but you claim that “a lot relevant literature is missing”. Could you provide some references that we failed to cite?
> * We have added the missing definitions of n and epsilon. Our mathematical statement is a simple rephrasing of the law of large numbers.
>
> We hope that you may consider reevaluating the paper in light these clarifications and additional experiments that we have performed.

---

> > ### Comment · Reviewer_ZCbx · 2022-12-07
> > **Response to Authors**
> >
> > The paper quality has been improved, and I have increased my score. But the method combines a few existing (or similar to existing) techniques together.

---

### Official Review · Reviewer_X8sx · 2022-10-23

**Confidence:** 4
**Correctness:** 2
**Technical Novelty And Significance:** 3
**Empirical Novelty And Significance:** 3
**Recommendation:** 8

**Clarity, Quality, Novelty And Reproducibility:**

Reproducibility: see question 3 below; there are questions about reproducibility as currently the reviewer thinks the improvements are needed.

Novelty and clarity: The paper seems novel, to the best of the reviewer's understanding, and the first three sections are really well written. On the description of the method and the experimental sections, the answers need to be given. The clarity of the method description needs to be improved (see comments below).

Questions on quality and clarity:

1. (Minor) Dimensionality: page 1, “Rather than compute costs which scale exponentially in the dimension, they instead typically have error” N must be defined

2. (Minor) Typos should be proofread e.g. “Page 2: These methods has proven successful “
 Also, should local in time solvers be spelled ‘local-in-time solvers’ as it is a compound adjective (see https://proofed.co.uk/writing-tips/a-guide-to-compound-adjectives-and-hyphenation/ for a better explanation of my point)? This question is totally subjective, and the reviewer intends it as an open question, which is in no way is reflected in my score.
3. While sections 1-3 give an excellent background,  it looks like section 4 does not give the method in one place in all its entirety; the reviewer also thinks that the method does not look reproducible from the description. Section 4.1 discusses a toy problem of a sum of two Gaussian like wave packets on which different  solvers are evaluated; is the proposed model a modified Du & Zaki (2021) architecture, or is it based on Bruna et al’s architecture? What is the compete list of modifications of this architecture?  It could be presented, for example, as some sort of algorithm. What are the hypoerparameters of the model?
4. In section 3, the description states that the computational complexity of the model is O(p^3 + p^2 n) and memory O(p^2 + pn); what is n and what is the exact architecture for which this assessment is made?
5. For the more complicated scenario, described in section 5, there need to be a comparison of models (with Bruna et al (2022) and Du& Zaki(2021)) and ablation studies: which of the proposed improvements actually make the impact on the solution: sinusoidal embedding, head fine-tuning etc? How does it work for different parameterisations of the equation? Up to what dimensionality is the solver still able to persist with stable learning, and at what dimensionality the proposed numerical scheme collapses? Is it possible to give more examples, e.g. some second-order PDEs or problems with the complex conditions, where the standard solvers fail (see Section 3 )


**Strength And Weaknesses:**

Strengths:
- The reviewer really enjoyed the problem statement in sections 1-3; it gives a good grasp on what the problem is and why the current (ML and non-ML based) solvers struggle to achieve good quality solutions for high dimensional problems, as well as complex initial conditions and for higher-order PDEs
- The problem is really interesting and in many aspects understudied

Weaknesses:
- Some of the details of the descriptions are failing to give the consistent picture of what the method looks like; see comments 3, 4
- Experimental assessment: the current experimental assessment lacks detail on why it is better than the existing solvers; see comment 5


**Summary Of The Paper:**

The paper describes a method which helps improve scalability of the local-in-time ODE solvers.


**Summary Of The Review:**

The reviewer thinks that while the introduction and the summary of the challenges of scalability of the local-in-time solvers are really clear and well written, the details of the proposed method and  be experimental assessment could be improved (see comments above).

==
Updating the scores to recommend acceptance as the authors, in my view, sufficiently addressed the comments

---

> ### Author Response · Authors · 2022-11-18
> **Author Response**
>
> We appreciate your detailed feedback and your emphasis on improving our method description; addressing these suggestions has improved the quality of the paper.
> Here is a list of changes that we have incorporated based on your review:
> * We added our method's pseudo-code in appendix D to show the method in its entirety (as well as the hyperparameters in appendix C). Also we are planning to share our python (JAX) codebase in Github.
> * We incorporated three comparisons to EDNN: wave maps (section 5), Vlasov (6+1) and Fokker-Plank (8+1) see updated Figure 3 and updated section 4.3 as well as appendix B. Also these last two equations are PDEs where standard solvers fail.
> * We integrated in the appendix an analysis of our method's PDE residual as we increase the dimensionality of the Fokker-Plank equation (in this equation we can naturally increase the spatial dimensionality), see Figure 5. We emphasize when the PDE residual starts to grow (even if it is <1e-1) it is unclear that the solution that we recover is of good precision and quality.
> * We included an ablation study on wave maps showing the benefits of the sinusoidal embedding, of head tuning and of scaling the method as seen in Figure 5 in the appendix. This is a really interesting result showing the relevance of the sinusoidal embedding, but also why scaling to larger nets and using more spatial samples makes a decisive difference.
>
> In regards to the questions that you raised:
> * Du & Zaki(2021) and Bruna et al (2022) are the same method except when the solution of the PDE is a probability distribution. In such a case, Bruna et al (2022) uses an active sampling scheme that improves the fit. In this respect, our method also follows the local-in-time approach proposed by Du & Zaki(2021) but we translate the framework into one that can be solved through Krylov subspace methods (by defining a fast MVM and then using CG with randomized Nystrom preconditioning). In terms of the neural network architecture, Du & Zaki (2021) use a small MLP whereas we first embed the input with the sinusoidal embedding and then pass it through an MLP with the wider layers as we can support larger nets.
> * For the computational cost, p is the number of parameters and n is the number of Monte Carlo samples taken in the spatial domain. Here the $O(p^3 + p^2n)$ cost depends on the model through the number of parameters and this is because, in contrast to our method, when forming a dense matrix $M = J^TJ/n$ the cost is $p^2n$ time, and a dense matrix solve consumes time $O(p^3)$
>
> We would appreciate if you would consider increasing your score in light of our response and general comments post.

---

> > ### Comment · Reviewer_X8sx · 2022-11-24
> > **Re: Author response**
> >
> > The rebuttal has significantly improved the paper, so I decided to increase the score and recommend acceptance:
> >
> > Comment 1) is solved
> >
> > Comment 2) All good, but I would suggest that in the final version "These methods has proven successful" should be corrected to "These methods have proven successful" (doesn't affect the score in neither iteration, just to make sure I flag this to authors, sometimes it's easy to overlook a couple of typos)
> >
> > Comment 3) Appendix D and hyperparameters in Appendix C sound all good to me. That answers the question of reproducibility, and looking forward to the code release after the decision
> >
> > Comment 4) Makes sense after the description improvement
> >
> > Comment 5) Figure 5 is really interesting indeed and is a good ablation study. If  I were very picky, I would suggest to change the colours to different ones on the left and the right, as the blue or green on the left is not the same as blue or green on the right, as standard inductive bias for the reader is that if colour is the same it's the same method (but it doesn't affect the score, just a suggestion to improve the presentation).

---

> > > ### Author Response · Authors · 2022-11-28
> > > **Updated with new feedback**
> > >
> > > Thank you for your engagement!
> > >
> > > * Sorry for overlooking the typo that you pointed before. We have already changed it.
> > > * We've changed the colors of the plots in Figure 5 to avoid creating a false connection between the plots.
> > >
> > > We tried updating the PDF with this feedback but it is not longer possible until the camera-ready window.

---

### Official Review · Reviewer_qtnF · 2022-10-24

**Confidence:** 3
**Clarity, Quality, Novelty And Reproducibility:** The paper is sufficiently clear and n…
**Correctness:** 4
**Technical Novelty And Significance:** 3
**Empirical Novelty And Significance:** 2
**Recommendation:** 8

**Strength And Weaknesses:**

Strength:

- Showing the deficiency of the PDE solvers.
- Suggesting new techniques to overcome the shortcoming of the PDE solvers.
- Increasing the scalability and conditioning of the PDE solvers.
- Proving (via theorems) why singular solutions happen (leading to ill conditioning) due to continuous parameter symmetry.

Weakness:

- Limited applications and examples.
- Lack of evaluation in high-dimensional PDE problems.

**Summary Of The Paper:**

This paper proposes a method, called Neural-IVP, for solving partial differential equations (PDEs) using neural networks. They use various helpful techniques to stabilize the PDE solutions, increase the scalability, and improve the representation power of the neural networks. Neural-IVP is a local in time method, meaning that the time dependency is induced by having time-variant parameters in the neural networks. They improve the representation power by using sinusoidal embedding and solving the last layer of the neural networks linearly. In order to increase the scalability and the number of parameters in the neural networks, they propose to use Jacobian-vector-products implemented using automatic differentiation, which makes the complexity of time and memory grow linearly by the number of parameters.

**Summary Of The Review:**

I believe that using neural networks to solve PDEs is an interesting and important problem to pursue. This paper has done a good job in showing the drawbacks of the previous methods for solving PDEs and has tried to address those by using various techniques. However, one main shortcoming of the paper is the lack of high-dimensional problems/examples. One of the selling points of this paper is the improved scalability. So, one expects to see its scalability to high-dimensional settings as well. In the section 5 of the paper, the title say "high-dimensional hyperbolic PDEs" but the paper only shows a (3+1) dimensional spacetime example. I am very curious to see if this method can work in hundreds-dimensional problems where neural ODE methods can work without problem? Or, if there is a limit in achievable dimensions, the authors should be clear about it. Also, reporting the training time versus dimension could also be helpful to see.

Minor comments:

There are some typos throughout the paper. These are some of them:

- be emploted to computer --> be employed to compute
- due the challenges --> due to the challenges

---

> ### Author Response · Authors · 2022-11-18
> **Author Response**
>
> We appreciate your constructive and thoughtful feedback. As you mentioned, we also believe that using neural networks to solve PDEs is an interesting and important problem, which we presume would garner future interest and applications.
>
> In our experiments we focused on second order in time PDEs as they tend to present a particular challenge for the Neural PDE solvers; however, we entirely agree that our paper would benefit from explicitly showing a broader set of applications and we have therefore included the 6+1 dimensional Vlasov equation and the 8+1 dimensional Fokker-Planck equation. On both of these equations we perform considerably better than the EDNN method, and finite difference methods are wholly impractical for the dimensions of these PDES.
>
> In terms of testing the limit of the method’s ability to scale to extremely large dimensions, we added an additional experiment and plotted the Fokker-Planck equation PDE residual error for our solver for 5, 10, and 20 dimensions as seen in Figure 5 in the appendix. As the dimension increases, so does the PDE solution error, and we expect that scaling beyond this amount will require a more sophisticated sampling strategy (as uniform becomes increasingly inefficient). We tried dimensions higher than 20 but the linear systems were not reaching small tolerances. Extending beyond 20 dimensions may require additional ideas.
>
> We appreciate if you would consider increasing your score in light of our response and separate general comments post.

---

> > ### Comment · Reviewer_qtnF · 2022-11-23
> > **Response**
> >
> > I would like to thank the authors for addressing my comments. Now that the authors show some high-dimensional PDE examples and acknowledge the limitation of their method to up to 20 dimensions, I think the paper is more clear. In light of their effort to improve the paper, I will increase my score.

---

> > > ### Author Response · Authors · 2022-11-28
> > > **On engagement**
> > >
> > > We appreciate your engagement and your continued support!

---

### Author Response · Authors · 2022-11-18
**General Response**

We thank all reviewers for their feedback. Inspired by reviewer comments, we modified the text and ran several experiments, which we now include in the updated version of the paper (available on the PDF link above, modifications are highlighted with orange text).

**On experimental assessment**. We focused our experiments on two types of problems that cannot be solved with current local-in-time methods: a second-order wave equation (section 4.3) and a second-order wave maps equation (section 5). However, we agree that adding a broader set of equations would strengthen the paper and therefore we have incorporated the Vlasov and Fokker-Planck equations. Note that, in all the comparisons, Neural-IVP has an order of magnitude lower PDE residual than EDNN (Du & Zaki 2021). The method from Bruna et al (2022) extends EDNN with adaptive sampling, but this approach is not directly applicable to these problems.

**On high-dimensionality**. Our goal is to provide a PDE solver that is mesh-free and that can tackle problems where current standard solvers struggle, such as the wave maps equation, or where the computational time is prohibitive such as the Vlasov equation (6+1) and our Fokker-Planck variant (8+1). Precisely in this context is that we call the previous PDEs high-dimensional, as the standard solvers suffer from the curse of dimensionality due to their dependence on grids. However, we do agree that in the context of ML it is common to think of high-dimensions as the space where images and text data belong. Therefore we have modified the text to avoid this confusion (please see changes in orange). Additionally, we have added an experiment that tracks the performance of our method as we increase the dimensionality as suggested by the reviewers. We use the Fokker-Planck equation as it is a natural example where we can incrementally increase the dimensionality of the problem, unlike the other problems that we consider in the paper. Nonetheless, we underscore that Neural-IVP is of considerable value for the dimensions that we have already considered and that our method should not be evaluated on a specific number of dimensionality that it can reach (as there are not open PDE problems of increasingly higher dimensions). Also the spatial dimension is not the only measurement of difficulty as the wave maps equation (4+1) is harder to solve than the Vlasov equation (6+1) or the Fokker-Planck variant (8+1) that we consider.

**Our method is the only neural PDE solver that can tackle a challenging second-order wave maps equation with realistic initial conditions**. Our method is able to do this due to the following key contributions:

(1) We identified three problems that prevent state-of-the-art Neural PDE methods from solving hard PDEs.

(2) By thinking deeply about network symmetries, representation power, and numerical conditioning, we were able to circumvented these problems and recast the neural PDE problem as one that can be solved efficiently via Krylov subspace methods even with many parameters.

(3) We demonstrated the empirical performance of our method, showing the clear advantages over both classical finite difference methods and neural PDE solvers.

We hope that our response and updates can be carefully considered.

---

### Decision · Program_Chairs · 2023-01-20

**Decision:**

Accept: poster

**Justification For Why Not Higher Score:**

The weaknesses listed above make this paper fall short of accept with spotlight.

**Justification For Why Not Lower Score:**

Paper merits acceptance given the reviews and the growing interest around using deep learning techniques to improve the state of the art in solving PDEs.

**Metareview: Summary, Strengths And Weaknesses:**

Summary: This paper is on the theme of developing better PDE solvers using deep learning techniques. The paper points out that techniques where a neural net globally parameterizes the spatio-temporal solution are susceptible to "catastrophic forgetting" as they use a single set of time-dependent weights. Using time-varying weights leads to the "local in-time" methods, but they are not very robust. The paper addresses these issues by developing a "Neural IVP" method that allows large capacity networks to be used while retaining stability and well conditioning.

Strengths: Proposals to improve PDE solvers are insightful: using fast matrix multiplies and the use of last-layer linear solves and sinusoidal embeddings improves representational capability; using regularization, restarts and last layer finetuning, instabilities are effectively mitigated.

Weaknesses: Restricted to low dimensional PDE solutions though claims are made on ability to handle "high dimensional" problems. Limited in experimental results - weak in terms of comparisons to other methods in terms of accuracy and speed.



**Note From Pc:**

if the above contains the word "oral" or "spotlight" please see: "oral" presentation means -> notable-top-5% and "spotlight" means -> notable-top-25%. As stated in our emails, we are disassociating presentation type from AC recommendations